# Invasive Burmese pythons alter host use and virus infection in the vector of a zoonotic virus

Nathan D. Burkett-Cadena [1✉], Erik M. Blosser[1], Anne A. Loggins[2], Monica C. Valente[3], Maureen T. Long[3], Lindsay P. Campbell[1], Lawrence E. Reeves [1], Irka Bargielowski[1] & Robert A. McCleery[2]

The composition of wildlife communities can have strong effects on transmission of zoonotic vector-borne pathogens, with more diverse communities often supporting lower infection prevalence in vectors (dilution effect). The introduced Burmese python, *Python bivittatus*, is eliminating large and medium-sized mammals throughout southern Florida, USA, impacting local communities and the ecology of zoonotic pathogens. We investigated invasive predator-mediated impacts on ecology of Everglades virus (EVEV), a zoonotic pathogen endemic to Florida that circulates in mosquito-rodent cycle. Using binomial generalized linear mixed effects models of field data at areas of high and low python densities, we show that increasing diversity of dilution host (non-rodent mammals) is associated with decreasing blood meals on amplifying hosts (cotton rats), and that increasing cotton rat host use is associated with increasing EVEV infection in vector mosquitoes. The Burmese python has caused a dramatic decrease in mammal diversity in southern Florida, which has shifted vector host use towards EVEV amplifying hosts (rodents), resulting in an indirect increase in EVEV infection prevalence in vector mosquitoes, putatively elevating human transmission risk. Our results indicate that an invasive predator can impact wildlife communities in ways that indirectly affect human health, highlighting the need for conserving biological diversity and natural communities.

[1] University of Florida, Florida Medical Entomology Laboratory, Vero Beach, FL, USA. [2] University of Florida, Wildlife Ecology and Conservation, Gainesville, FL, USA. [3] University of Florida, Department of Comparative, Diagnostic & Population Medicine, Gainesville, FL, USA. ✉email: nburkettcadena@ufl.edu

The indirect or cascading impacts of invasive species are difficult to quantify and predict[1–3], yet have the potential to be even more damaging and costly than the direct impacts[3,4]. Particularly worrisome are the indirect impacts that invasive species may have on human disease[5,6], given the profound ways that invasive species shape ecosystems and the importance of these ecosystems in supporting or suppressing the transmission of pathogens[7,8]. One potential pathway that invasive species alter human disease risk is by impacting the structure of vertebrate communities[5,8]. In a given community, individual vertebrate species (both wild and domestic) contribute unequally to the amplification of vector-borne pathogens, such that a few key host species drive amplification and spillover[9,10]. Other animals that are fed upon by the vector, but are poor hosts of the pathogen (dilution hosts), can lower pathogen prevalence in the community, via a "dilution effect"[11]. Any number of factors can cause changes to the vertebrate community (e.g., land-use change, climate change, and invasive species), with cascading impacts for vector-borne disease risk. The "perturbation hypothesis"[12] suggests that pathogen spillover is caused by human-induced disturbances to the ecosystem, through shifts in inter-species transmission and rates of pathogen prevalence. Thus, the composition of the vertebrate community affects interactions between vectors and their host animals, leading to indirect impacts on human risk of infection.

The invasive Burmese python, *Python bivittatus*, is a major perturbation to the vertebrate community and ecosystem of southern Florida (USA), and has been incriminated in precipitous declines in native mammals throughout southernmost Florida[13–16], with an 85–100% decrease in the frequency of observations of raccoon, opossum, bobcat, and rabbits[13]. The loss of mammal diversity is thought to be causing a complete restructuring of the food web, declines in ecosystem function, and an array of cascading ecological effects[14,17], such as increased predation on nests of oviparous animals[17]. In the Florida Everglades, the decline in medium and large-sized mammals, attributed to the establishment of the Burmese python in the 1990s, has resulted in a major shift in host use by the vector of a human pathogen[18]. Large and medium-sized mammals (deer, raccoon, and others) constituted less than 2% of vector bloodmeals in 2016, compared to 50% in the 1970s, prior to the invasion and establishment of the Burmese python[18,19]. The dramatic decrease in feedings on medium- and large-sized mammals over this interval was offset by a striking increase in feedings on rodents[18,19], particularly the hispid cotton rat, *Sigmodon hispidus* (from 14.7% in 1979 to 76.8% in 2016). Hispid cotton rat is the primary host of Everglades virus[20–24] a mosquito-borne virus endemic in south Florida that can cause clinical encephalitis in humans[25] and represents subtype II of the Venezuelan equine encephalitis virus complex. Recent work in the northern Everglades suggests that rodent communities have changed compared with historical data, with the hispid cotton rat now the dominant rodent species[26]. Shifts towards increased feedings on these amplifying hosts, mediated by Burmese python predation, could increase the prevalence of Everglades virus in vectors, potentially leading to increased risk of human infection in affected areas. However, this pattern of increased vector infection and transmission has not yet been demonstrated.

In the current study, we investigated the potential indirect impacts of an invasive predator, the Burmese python, on EVEV prevalence in the mosquito *Culex cedecei*, the only confirmed vector of EVEV[19,27,28]. Working in ecologically similar areas in southern Florida with historically comparable mammal communities, we quantified mammal diversity and activity, along with vector host use and virus infection in areas that differ in the relative presence of Burmese python (Fig. 1). In order to understand the relationship between vector infection rate and the mammal community, we developed models to characterize the association between host blood meal ratios (cotton rat blood meals/total blood meals) and vector infection rates with metrics of mammal activity and diversity. In addition, we attempted to identify vertebrate species that may serve as dilution hosts in this virus system by providing a relatively large fraction of blood meals for vectors, occurring at relatively high densities, and having low reservoir competence.

## Results

Blood meals from cotton rats, the confirmed host of EVEV, varied substantially among sites, constituting between 0 to 63.4% of total blood meals among sites with ten or more blood meals (Fig. 2a, b). Although uncertainty was high at model extremes, the model predicted estimates suggest that relative cotton rat host use increases approximately fivefold across the recorded range of cotton rat activity (Fig. 2a), and decreases by ~90% across the range of non-rodent diversity (Fig. 2b). Using binomial generalized linear mixed models (GLMMs) to quantify the relationships between relative cotton rat host use (ratio of cotton rat blood meals: all other blood meals) and metrics describing the composition of mammal community (cotton rat activity, non-rodent activity, and non-rodent diversity), our most parsimonious model (AICc weight = 0.592; Supplementary Table 1) included the fixed effects of cotton rat activity and non-rodent diversity. Both variables were relevant predictors of cotton rat host use (Table 1). We found lesser support for two models with one fixed effect (non-rodent activity, cotton rat activity, Supplementary Table 1) and a model with the fixed effects of non-rodent diversity and non-rodent activity (Supplementary Table 1). However, like our most parsimonious model, cotton rat activity and non-rodent diversity show relatively strong positive and negative relationships with cotton rat host use and were the only relevant predictors (Table 1). We weighted the relative host use by the number of blood meals obtained at each sampling site, included a random effect by the site to account for overdispersion in proportional data, and used an information-theoretic approach to identify the most parsimonious models (i.e., Δ AICc and AICc weights). We considered variables within these models to be relevant predictors of cotton rat host use if their 95% CI of their beta estimates did not include 0 and their Wald test $p$ values < 0.05. We assessed the strength of relevant variables by graphic model-based predictions. Full model summaries including β estimates, standard errors, $p$ values, and 95% CI are available in Supplementary Table 1.

The vector infection rate varied from low (EVEV not detected at four sites) to quite high maximum likelihood estimates (MLE = 3.24), even between some sites separated by <2 km (Fig. 1 and Table 2). To explain this variation, we used binomial GLMMs to quantify the relationships between EVEV infection rate (MLE) in the vector and cotton rat activity, non-rodent activity, non-rodent diversity, and the relative cotton rat host use. We weighted the proportion of EVEV positive pools by the total number of pools sampled at each site, and include a site-level random effect to account for overdispersion. Evaluating model parsimony, we found that most of our models contributed, at least marginally, to predicting EVEV infection rates (Table 1). However, within our models only one variable, relative cotton rat host use, found in the three most parsimonious models, was a relevant predictor of EVEV infection rates (Supplementary Table 1). Based on the predictions from our most parsimonious model, we found that as relative cotton rat host use increased from 0 to >0.50 of blood meals, EVEV infection rates increased approximately threefold (Fig. 2c). High uncertainty in EVEV

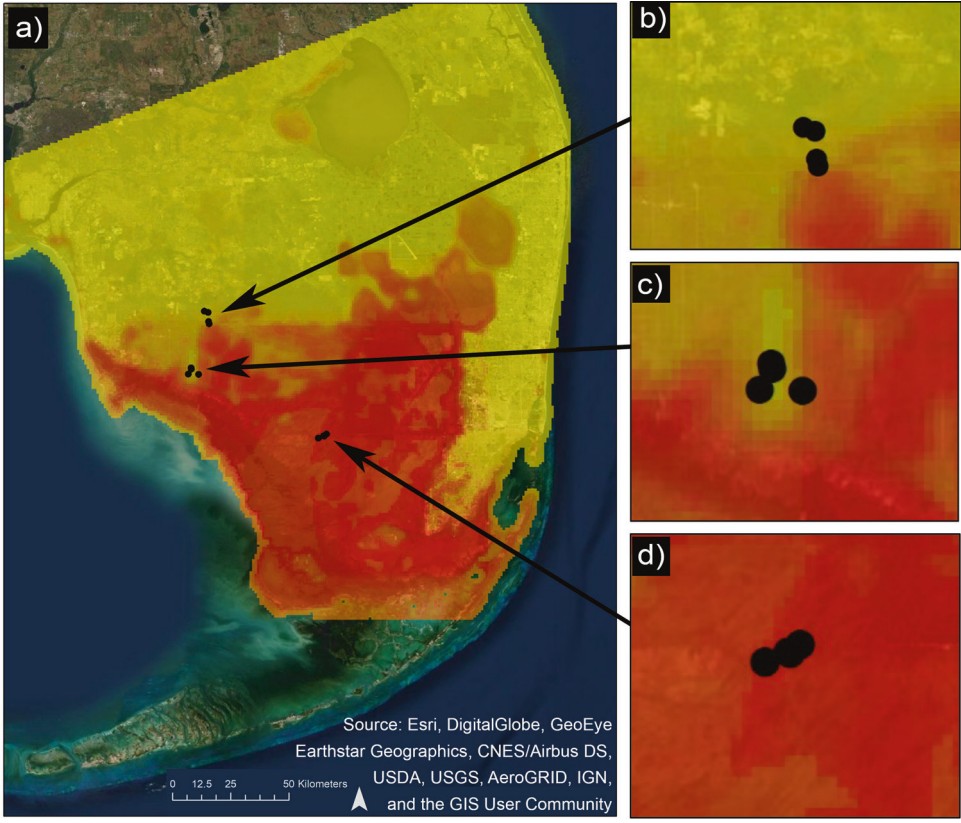

**Fig. 1 Map of southern Florida with modeled relative presence of Burmese python and study sites.** Burmese python observations were modeled using 1165 georeferenced python locations (2014 to 2017) from the Early Detection and Distribution Mapping System data base (https://www.eddmaps.org/). Shades of red indicated a higher estimated relative python presence. Relative python presence values generated by species distribution modeling using an ensemble modeling approach that combined model outputs from multiple SDM algorithms, executed in the "biomod2" package in R. Black circles indicate mammal and mosquito sampling locations in southern Florida (**a**) at Florida Panther National Wildlife Refuge (**b**), Fakahatchee Strand Preserve State Park (**c**), and Big Cypress National Preserve (**d**).

infection rates was observed at model extremes (Fig. 2c, d), and only one site had very high cotton rat host use (63.6%) and high EVEV infection rate (3.2/1000), which likely influenced model outcomes.

Examining model residuals from our most parsimonious model from each predictor (relative cotton rat host use and EVEV infection rate) we found no indication of spatial autocorrelation (Supplementary Fig. 1). Additionally, plotting model residuals against longitude showed no evidence of high or low clustering in residual values was found (Supplementary Fig. 1).

To identify potential EVEV dilution hosts and explore how Burmese pythons may impact EVEV transmission through effects on the mammal community we plotted non-rodent activity (including gray squirrel) and host use across estimates of python presence, generated through a species distribution model output using georeferenced python observations and environmental data, using an ensemble modeling approach. Non-rodent activity and host use of *Cx. cedecei* differed greatly across sites, which spanned a wide range of relative probability of Burmese python presence (Fig. 3a–d). Non-rodent activity (and species richness) decreased dramatically as relative probabilities of python presence increased (Fig. 3a). At the three sites with the highest relative probabilities of python presence (all located within Big Cypress Preserve) both mean (Fig. 3a) and cumulative (Fig. 3b) non-rodent activity was consistently low (0.008–0.013). The non-rodent activity was between 2.8 and 14.0 times higher (mean = 9.1) at sites ($n = 7$) with a lower relative probability of python presence values (Fig. 3a). The activity of non-rodents decreased as the relative

probability of python presence values increased (Fig. 3b), while rodent activity was robust at all sites (Fig. 3c).

The reduction in non-rodent activity with an increasing relative probability of python presence coincided with shifting patterns of host use of the vector (Fig. 3d). In general, *Cx. cedecei* fed upon a wide variety of available mammals (6 orders, 15 species) across sites including diverse non-rodents (Fig. 3e) and four muroid rodent species (Fig. 3f). While the mammal species bitten varied considerably among sites (Fig. 3d), relative cotton rat host use generally increased with an increasing relative probability of python presence (Fig. 3c). Host affinity (ratio of relative host use to relative abundance) changed with respect to the relative probability of python presence for the most commonly bitten non-rodent species (Fig. 4). We found evidence that two mammals, marsh rabbit white-tailed deer, were preferred by *Cx. cedecei* (Fig. 4) at sites with low and moderate python presence, respectively, indicating that they are potentially important dilution hosts. These two putative dilution hosts species were both absent from sites with high relative probability of python presence values (Figs. 3b, 4). Eastern gray squirrel, in contrast, was present at all sites and composed an increasingly high proportion of the mammal community as the relative probability of python presence increased, yet this species contributed only minimally (<10%) to the blood meals of *Cx. cedecei*. Mammal species bitten by *Cx. cedecei* included a marsupial (Virginia opossum), a xenarthran (nine-banded armadillo), multiple carnivorans (black bear, Florida panther, Florida mink, raccoon), an ungulate (white-tailed deer), a lagomorph (marsh rabbit), muroid rodents (cotton

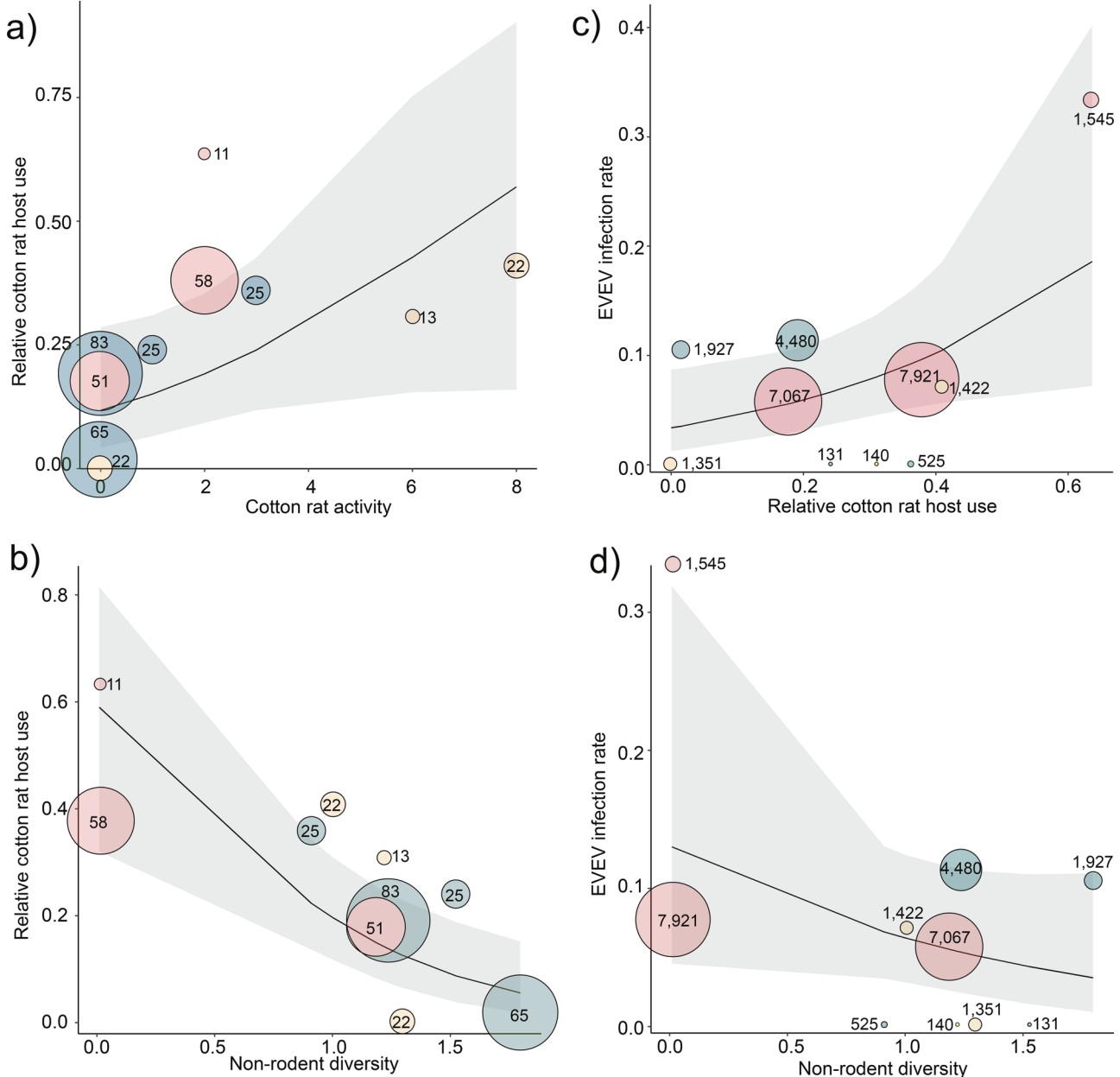

**Fig. 2 Variable effect plots from GLMM models of mammal community metric, host use, and Everglades virus infection.** Effects of **a** Cotton rat activity and **b** Non-rodent diversity on relative cotton rat host use. Effects of **c** Relative cotton rat host use and **d** Non-rodent diversity on a proportion of EVEV positive mosquito pools. Shaded areas represent 95% confidence intervals. The size of each point represents the number of total blood meals (**a**, **b**) or the number of females screened by PCR. Model covariates were weighted by sampled size and a site-level random effect was included to account for overdispersion. The colors of points represent high (red), moderate (orange), or low (blue) estimated python presence, based upon estimated relative presence values from the species distribution model output.

rat, cotton mouse, black rat, house mouse), and one non-muroid rodent (gray squirrel). Raw blood meal and virus infection data were provided in Supplementary Data 1 and 2, respectively.

## Discussion
Our findings that cotton rat activity and non-rodent diversity were the best predictors of relative cotton rat host use, and that relative cotton rat host use was the best predictor of EVEV infection rates in the vector suggests that an invasive predator can alter host communities in ways that indirectly alter the risk of vector-borne pathogen transmission. Importantly, the data supported a strong negative effect of non-rodent diversity on vector

host use (ratio of blood meals from cotton rat versus other mammals), but the data supported a minimal and small negative effect of non-rodent diversity on virus infection rate in the vector. This indicates that changes to the host community do not directly impact infection rate in vectors, but changes in the host community drive host use, which has indirect impacts on the infection rate in the vector.

The finding that the proportions of blood meals from cotton rats increased as large and medium-sized mammal activity decreased (Fig. 2a) aligns with our previous work[18] demonstrating that vectors feed more heavily upon cotton rats in areas with python-impacted vertebrate communities, compared to those same areas before the python invasion. Our finding that measures

**Table 1 Rankings of models used to explain the variation in the mammal community, host use, and Everglades virus (EVEV) infection rate in the vector.**

| Models | Residual DF | -2LL | AICc | ΔAIC | AICc wt | $R^2$ |
|---|---|---|---|---|---|---|
| **Relative cotton rat host use** | | | | | | |
| Cotton rat activity + Non-rodent diversity | 6 | −25.09 | 66.19 | 0.00 | 0.59 | 0.34 |
| Non-rodent activity | 7 | −30.34 | 68.40 | 2.21 | 0.20 | 0.39 |
| Cotton rat activity | 7 | −29.58 | 69.16 | 2.98 | 0.13 | 0.39 |
| Non-rodent activity + Non-rodent diversity | 6 | −27.33 | 70.66 | 4.47 | 0.06 | 0.35 |
| Cotton rat activity + Non-rodent activity | 6 | −28.77 | 73.54 | 7.36 | 0.01 | 0.39 |
| Non-rodent diversity | 7 | −34.55 | 79.09 | 12.91 | 0.00 | 0.35 |
| **EVEV Infection rate** | | | | | | |
| Cotton rat activity + Relative Cotton rat host use | 6 | −24.41 | 58.53 | 0.00 | 0.28 | 0.17 |
| Relative Cotton rat host use | 7 | −25.81 | 59.33 | 0.80 | 0.19 | 0.23 |
| Cotton rat activity + Relative cotton rat host use + Non-rodent diversity | 5 | −24.40 | 60.52 | 1.99 | 0.10 | 0.15 |
| Non-rodent activity | 7 | −26.50 | 60.74 | 2.21 | 0.09 | 0.27 |
| Non-rodent diversity | 7 | −26.51 | 60.74 | 2.21 | 0.09 | 0.22 |
| Non-rodent activity + Relative Cotton rat host use | 6 | −25.71 | 61.14 | 2.61 | 0.08 | 0.20 |
| Non-rodent activity + Non-rodent diversity | 6 | −26.20 | 62.12 | 3.60 | 0.05 | 0.25 |
| Cotton rat activity + Non-rodent activity | 6 | −26.34 | 62.40 | 3.87 | 0.04 | 0.28 |
| Cotton rat activity + Non-rodent diversity | 6 | −26.35 | 62.41 | 3.88 | 0.04 | 0.22 |
| Cotton rat activity | 7 | −27.50 | 62.72 | 4.19 | 0.03 | 0.23 |

GLMM models were evaluated based on their residual degrees of freedom (DF), −2 log-likelihood (−2LL), AICc, Change (Δ) in AICc, AICc weight (AICcwt), and pseudo $R^2$.

of the non-rodent diversity were a strong predictor of blood meal composition in the vector (Fig. 1b) has important implications for understanding the links between community diversity and disease. The ratio of blood meals from cotton rats to other mammals was inversely linked to non-rodent diversity, demonstrating that the presence, and by extension, the loss, of these animal species impact the patterns of host use of the vector and supports the assertion that dilution hosts are important drivers of contact between vectors and amplifying hosts and, as a consequence, the infection rate in the vector.

EVEV infection rates in the vector increased with relative cotton rat host use. A similar complex relationship between pathogen prevalence, host abundance, and measures of host use has been observed in other vector-borne pathogen systems, notably Lyme disease (*Borrelia burgdorferi*). The prevalence of Lyme spirochetes in vector ticks was significantly higher in the northern than the southern US, a pattern attributed to selective feeding on noncompetent hosts (lizards) in the southern US (Ginsberg et al. 2021). Host (mouse) abundance was negatively associated with ticks per host animal, which affected the distribution of ticks per animal. The host community available to the vector drives patterns of host use. As the diversity of potential vector host animals increases, infection prevalence in vector decreases[9,29]. Lower mammal diversity can result from disturbances to otherwise natural ecosystems, such as forest fragmentation[30,31]. Here we show that disruptions to mammal communities caused by an invasive predator cascade through the system, altering vector–host associations and increasing vector infection rates of a zoonotic disease. We find that diversity within the mammal community has a strong effect on patterns of host use, with the availability of non-rodent (dilution) hosts in the environment impacting vector feedings on competent virus hosts (cotton rats). Diversity, in turn, appears to have a negative indirect effect on the vector infection rate.

The low rodent and non-rodent activity observed at sites with the greatest relative probability of python presence values (Fig. 2a) was startling but is supported by previous studies in southern Florida that have documented precipitous declines in python-invaded areas[13,14,16]. While the lower activity of large and medium-sized mammals was expected in areas with a higher relative probability of python presence, the lower rodent numbers

from our trapping in these same areas were not expected. Previous work using the unconventional method of sighting rodents from a vehicle suggested that rodent abundance increased slightly after the establishment of Burmese pythons in Everglades National Park[13]. This change was attributed to their high reproductive potential and severe declines in their predators: bobcat (*Lynx rufus*) and foxes (*Urocyon cinereoargenteus* and *Vulpes vulpes*)[13]. Nonetheless, rodents continued to contribute a large number and proportion (Fig. 3b, d) of blood meals to *Cx. cedecei* in locations with an elevated relative probability of python presence values, indicating that vectors continue to encounter sufficient rodent hosts to sustain relatively high levels of EVEV transmission (Fig. 4c, f).

Interestingly, the highest infection rate (six EVEV positive samples from 1545 *Cx. cedecei* females; MLE = 3.24) was observed at a site (Tree Snail Nature Trail) with the lowest numbers of total mammal detections (rodents and non-rodents). At this site, only three mammal species were observed in mammal surveys (hispid cotton rat, cotton mouse, and gray squirrel) during sampling. No medium-sized or large mammals were observed in mammal surveys or blood meals at this site (Fig. 3e and Supplementary Data 1). Low numbers of blood-engorged mosquitoes were collected at this site (n = 11), compared to some other sites, yet 100% of blood meals were from muroid rodents (cotton rat, cotton mouse, and black rat). This suggests that the absence of non-rodents concentrates feedings on virus hosts, which results in overall higher virus infection rates.

Identifying dilution hosts, species which serve to lower overall infection prevalence by providing a large fraction of blood meals for vectors, occurring at relatively high densities and having low reservoir competence[29], is an important conservation goal with repercussions for disease ecology. Marsh rabbit and white-tailed deer were found to be selected by *Cx. cedecei* (Fig. 4) where relative probabilities of python presence were low or moderate, respectively. The relatively large numbers of deer and rabbits at these sites (Fig. 3b) corresponded to low cotton rat host use (Fig. 3d) and could partially explain the absence of detected virus at some sites. Our analysis suggests that these mammal species should serve as important dilution hosts for EVEV in areas where they persist. Unfortunately, white-tailed deer, marsh rabbit, and other medium and large mammals have already been heavily

**Table 2 Sampling locations, estimated python presence, cotton rat activity, host use, and virus infection.**

| Property name | Site code | Site name | Estimated python presence | Coordinates | Cotton rat activity | Non-rodent activity | Blood meals | Cotton rat blood meals | Vectors screened | EVEV positive pools |
|---|---|---|---|---|---|---|---|---|---|---|
| Big Cypress | BC1 | Pinecrest | 933 | 25°45′41.77′′N, 80°55′9.22′′W | 2 | 0.013 | 58 | 22 | 7921 | 6 |
| Big Cypress | BC2 | Mitchell Landing 2 | 928 | 25°45′18.55′′N, 80°55′44.92′′W | 0 | 0.008 | 2 | 0 | 3197 | 0 |
| Big Cypress | BC3 | Mitchell Landing 1 | 933 | 25°45′14.40′′N, 80°55′36.74′′W | 2 | 0.013 | 51 | 9 | 7067 | 4 |
| Big Cypress | BC4 | Tree snail | 956 | 25°44′48.48′′N, 80°56′53.69′′W | 2 | 0.010 | 11 | 7 | 1545 | 5 |
| Fakahatchee | FAK1 | Gate 2 | 386 | 25°58′42.05′′N, 81°23′4.77′′W | 6 | 0.180 | 3 | 3 | 65 | 0 |
| Fakahatchee | FAK2 | Gate 7 | 326 | 25°58′45.80′′N, 81°25′19.68′′W | 0 | 0.135 | 22 | 0 | 1351 | 0 |
| Fakahatchee | FAK3 | Gate 8 | 123 | 25°59′49.2′′N, 81°24′42.1′′W | 0 | 0.140 | 83 | 16 | 4480 | 5 |
| Fakahatchee | FAK4 | Gate 9 | 119 | 26°00′05.6′′N, 81°24′43.0′′W | 0 | 0.053 | 65 | 1 | 1927 | 2 |
| Florida Panther | PAN1 | Work Center | 123 | 26°0′16.17′′N, 81°20′54.66′′W | 1 | 0.103 | 25 | 4 | 131 | 0 |
| Florida Panther | PAN2 | Hunt 2 | 112 | 26°12′13.83′′N, 81°21′1.27′′W | 3 | 0.056 | 25 | 9 | 525 | 0 |
| Florida Panther | PAN3 | Hunt 3 | 352 | 26°12′28.35′′N, 81°21′48.49′′W | 8 | 0.028 | 22 | 9 | 1422 | 1 |
| Florida Panther | PAN4 | Hiking trail | 261 | 26° 9′44.49′′N, 81°20′46.88′′W | 6 | 0.123 | 13 | 6 | 140 | 0 |

Relative Python presence values were generated using species distribution models. Cotton rat activity was quantified using stratified Sherman trapping. Non rodent activity was quantified using wildlife cameras along trails. Blood meals were determined using PCR-based assays targeting vertebrate DNA in blood-engorged mosquitoes. Everglades virus infection in vector mosquitoes was determined using RT-PCR assays targeting non-structural proteins.

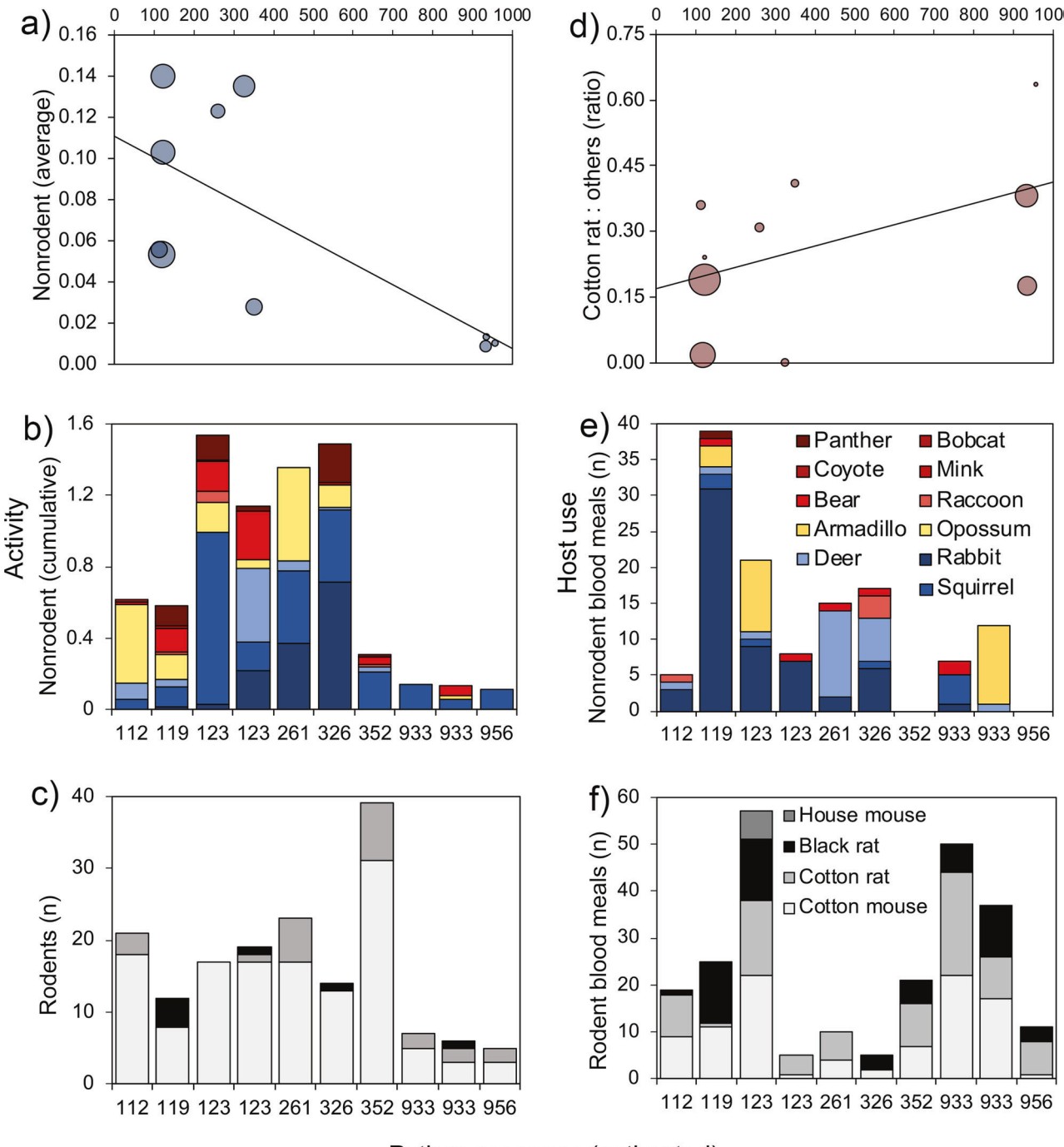

**Fig. 3 Mammal activity and host use at sites with varying relative probability of Burmese python presence.** Average (**a**) and cumulative (**b**) non-rodent activity was quantified through camera trapping along corridors. Rodent activity (**c**) was quantified using Sherman traps. Host use as a function of the proportion of blood meals from cotton rats to other mammals (**d**), non-rodents (**e**), and rodents (**f**) was determined through PCR-based analysis of blood-engorged *Culex cedecei* females. Values of the relative probability of python presence were generated by a species distribution model using an ensemble modeling approach that combined model outputs from multiple SDM algorithms, executed in the "biomod2" package in R. Points in (**a**) and (**c**) are scaled by total activity and numbers of blood meals, respectively.

impacted by Burmese python in southern Florida[13,14,17], reducing their potential to serve as dilution hosts in these systems, without restorative actions. Interestingly, urbanization has been found to somewhat limit impacts of Burmese python on raccoon and marsh rabbits, by increasing availability of their food and lowering densities of their natural predators[16]. The resilience of raccoon and marsh rabbits in peri-urban environments could help to suppress the human risk of EVEV in these areas.

This study has several limitations. Our inferences of relative probabilities of python presence were modeled from citizen science data (EDDMapS), which could be biased by user inputs, variation in sampling effort, and sampling from roads[31]. To address this issue we accounted for sampling bias using a bias correction approach by which background points used in the model calibration were generated with a similar bias to the citizen science python occurrence data, mitigating patterns that may

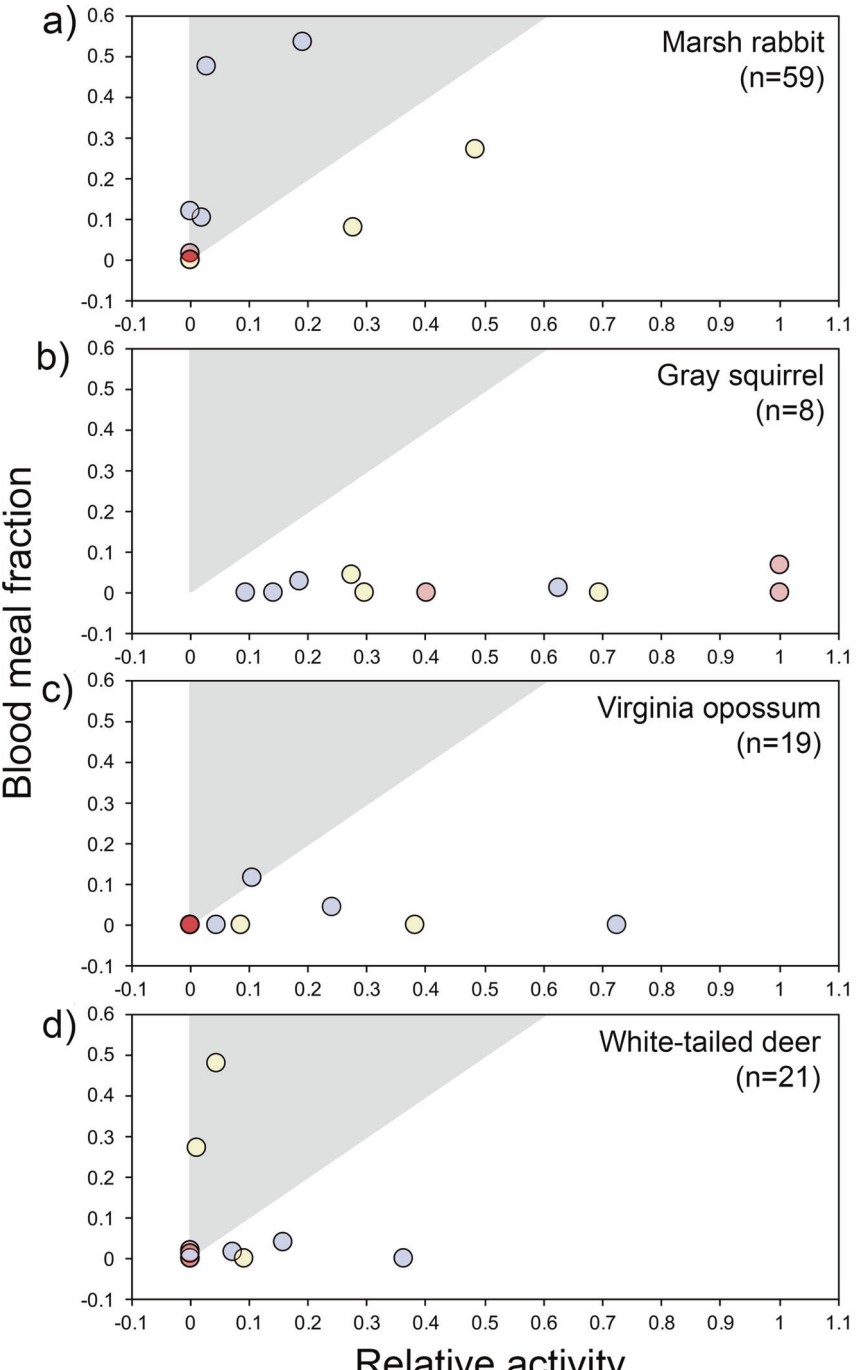

**Fig. 4 Relative activity and blood meal fraction of mammals across sites with varying relative probabilities of python presence.** Total numbers of blood meals from marsh rabbit (**a**), gray squirrel (**b**), Virginia opossum (**c**), and white-tailed deer (**d**) are provided in each panel. Colors of points represent high (red), moderate (yellow), or low (blue) estimated python presence, based upon species distribution model output. Gray triangular shading of the plot area indicates host preference (ratio of host use to relative abundance) by *Culex cedecei*. The relative activity of mammals was quantified through stratified camera trapping. The blood meal fraction was quantified using PCR-based assays to identify the vertebrate source of DNA in blood-engorged female mosquitoes.

result from opportunistic observations close to recreational trails, observation towers, or roads (see details in Methods). Nonetheless, with no rigorous sampling methods for invasive pythons yet available, our approach uses the best available data and should provide a robust approximation of Burmese python activity in the study areas[32]. It is also important to note that our measures of infection prevalence in the vector were estimated based upon variable sample sizes of *Cx. cedecei*, with low numbers at some

sites. The average infection rate at sites where the virus was detected (six sites) was 1.24 per 1000 females, so the actual infection rates at sites with fewer than 1000 females screened have large confidence intervals. Our analysis compensated for this variability by weighting MLE by the numbers of mosquito pools screened for EVEV at each site. Finally, the higher observed EVEV infection prevalence in the vector mosquito is not necessarily associated with higher disease incidence (or risk) in

humans, because human disease cases are dependent on human contact with infected vectors and human contact with vectors is not necessarily driven by the same conditions that produce high vector infection prevalence[30]. Future studies might gain a better estimate of human risk using sentinel rodents and sampling from a greater number of locations, and by including additional variables which are known to influence animal communities (flooding, fire, invasive plants, etc.).

## Methods

**Field sites**. We sampled mosquitoes and mammals at 12 sites across southern Florida, spanning a range of estimated invasive python densities. At each site we trapped mosquitoes and mammals and used the samples to interrogate hypotheses regarding the relationships between host communities, vector host use, and Everglades infection, as they relate to impacts of the invasive Burmese python. Twelve sites were established, divided evenly among Big Cypress National Preserve, Fakahatchee Strand Preserve State Park, and Florida Panther National Wildlife Refuge. Sites were spaced >300 m apart in hardwood hammock vegetation habitats, surrounded by various freshwater wetlands (primarily cypress swamp, freshwater marsh, or cypress strand) (Fig. 1).

**Mosquito collections**. We used carbon dioxide baited CDC miniature light traps[33] and pop-up resting shelters[34], to sample *Cx. cedecei*. Battery-powered CDC mini light trap with incandescent light (Model 2836BQ; BioQuip Products, Rancho Dominguez, CA, USA) were baited with dry ice (~0.5 kg) and hung from tree limbs at ~1.5 m above ground level. Traps were set prior to dusk and allowed to run overnight, until retrieval the following morning, for 78 trap nights. Nine resting shelters were placed at each of twelve sampling sites for each sampling period. A battery-powered aspirator, constructed of a handheld vacuum (BDH1800S Ni-Cd 18 V Hand Vac, Black & Decker, Maryland, USA), modified to use mesh-bottom collection canisters (Model 2846D; BioQuip; Rancho Dominguez, CA, USA) was used to sample resting adult mosquitoes from the shelters. We conducted mosquito sampling on ten separate occasions in September and October 2017, with 3–6 sampling days per location.

**Mammal community**. We quantified mammal communities at all 12 sites using cameras and live trapping. We sampled each site for 3 weeks (1 week per month) with three digital trap cameras (Bushnell Trophy Camera, Overland Park, KS) placed in areas of potential mammal activity (i.e., trails). We adjusted camera detections to reflect the number of independent visits by each species in a 20-min increment[35]. Additionally, at each site we placed 20 H. B. Sherman folding traps (3 × 3.5 × 9″, H. B. Sherman Traps, Inc., Tallahassee, FL, USA) in two parallel ten-trap transects with 10 m spacing. We opened these traps for four consecutive nights every month for 3 months (12 days total). We identified all rodent species and fitted each individual with an ear-tag (1005-1, National Band Co., Newport, KY, USA). Our protocols were approved by the University of Florida Institution Animal Use and Care Committee (#201709906). Due to relatively low capture rates, we generated estimates of rodent activity based on the number of unique individuals (i.e., minimum number alive estimates[36]). Measures of rodent and non-rodent host diversity and evenness were calculated from independent visits and unique individuals using Hill numbers in the "vegan" package in R[37].

**Molecular assays**. Blood-engorged *Cx. cedecei* females were processed individually for host source using PCR-based blood meal analysis, followed by Sanger sequencing comparison with GenBank databases. In brief, total genomic DNA was extracted from individual blood-engorged females using InstaGene Matrix (Bio-Rad, USA) followed by three PCR assays (Supplementary Information) targeting conserved mitochondrial (cytochrome b) and ribosomal (16 s) genes[18,33,38] of

mammals, amphibians, birds, and reptiles. Sequencing (forward direction) was performed by Eurofins Genomics (Louisville, KY, USA) and sequences were compared to available vertebrate sequences in GenBank using the BLASTn function. Sequences with >95% similarity to GenBank sequences were considered a positive match.

We screened RNA extracts of pooled *Cx. cedecei* (25 or fewer nonengorged females) homogenate for alphaviral RNA using probe and primers of a pan-alphavirus TaqMan reverse transcriptase quantitative PCR (RT-qPCR) assay targeting non-structural protein 4 (Supplementary Information), following protocols in ref. [39]. To confirm positive results, we sequenced amplicons from all Alphavirus-positive RT-PCR samples using two-step conventional RT-PCR (Supplementary Information). We calculated the vector infection rate at each location using MLE based on probabilistic models following a binomial distribution that can be adapted for use with variable pool sizes[40–43].

**Statistics and reproducibility**. To produce a relative probability of Burmese python presence values across the region and at our study sites, we downloaded 1165 georeferenced python observations collected during a 4-year period, prior to the year of the field study (2014 to 2017) from the Early Detection and Distribution Mapping Systems (EDDMapS) repository (https://www.eddmaps.org). We generated a species distribution model using python occurrence records as presence data, and urban land cover, home range values, and latitude and longitude raster data at a 1 km spatial resolution served as environmental variables. Urban landcover was acquired using the National GAP Analysis Program Landcover Data[44]. Python home range was were generated at the 4-km scale, corresponding to the home range size of pythons in ENP[45]. Sampling bias is a common phenomenon in presence-only data sets because of a greater number of occurrences reported near roadways or other easily accessible areas, which can impact model results. Here, we implemented a bias-correction approach to model calibration, generating a set of background data points exhibiting a similar bias to the occurrence data obtained from the EDDMapS repository, restricting background points within 100 m of a road. This procedure ameliorates the impacts of sampling bias, leading to increased model accuracy and model predictions[46,47]. The species distribution model was generated using an ensemble modeling approach that combined model outputs from nine SDM algorithms, executed in the "biomod2" package in R using default settings. The mean of the ensemble output was used as our final model output with values ranging between 0 and 1000, with values closer to 1000 indicating a greater relative probability of python presence (Fig. 1). Model evaluation statistics and plotted occurrence data are available in Supplementary Figs. 1 and 2, respectively. We extracted the relative probability of presence values at study site locations and plotted these values with biological variables to compare values visually across the sample locations.

Due to relatively low capture rates, we generated estimates of rodent activity based on the number of unique individuals (i.e., Minimum Number Alive Estimates[36]). Measures of non-rodent diversity (including squirrels) were calculated from independent visits and unique individuals using Hill numbers in the "vegan" package in R[37]. We quantified relationships between vector infection rates, relative cotton rat host use (ratio cotton rat blood meals: all other blood meals), cotton rat activity, non-rodent activity, and diversity (excluding muroid rodents) (Table 3) using two sets of binomial generalized linear mixed effect models (GLMMs) that included a site-level random effect to account for overdispersion. Relative cotton rat host use, weighted by the total number of blood meals at a location, served as the response variable in the first candidate set. The proportion of EVEV positive mosquito pools, weighted by the total number of pools tested at a location, served as the response variable in the second candidate set. A Pearson's correlation matrix (Supplementary Table 2) identified the potential for multicollinearity between predictor variables, and we included only variables with $r < ±0.6$ in the same candidate model. We evaluated models based on an information-theoretic approach using Akaike's Information Criterion, corrected for small sample size (AICc)[48,49]. We ranked models based on their parsimony (i.e., lowest AICc) before calculating AICc weights (AICc$_w$). We evaluated variables within our models and considered them to be relevant predictors if their 95% CI of

---

**Table 3 Metrics of mammal activity used in the analysis.**

| Metric of mammal activity | Formulation of metrics | Biological relevance |
|---|---|---|
| Cotton rat activity | Number of unique cotton rats captured per site | Cotton rat is the only laboratory-confirmed vertebrate host of EVEV. |
| Non-rodent diversity | Shannon index of non-rodent activity | Greater diversity of non-rodents may reduce rodent blood meals and EVEV prevalence. |
| Non-rodent activity | Average non-rodent activity (number of independent pictures of each species/number of days cameras were active) | Higher activity of non-rodents may reduce rodent blood meals and EVEV prevalence. |
| Relative cotton rat host use | Ratio of blood meals from cotton rats to all other animals. | Cotton rats are the only confirmed natural host for which all host competence criteria are satisfied. |

Explanations of the formulation and relevance of derived from camera and live trapping data from Greater Everglades Ecosystem in south Florida.

their β estimates did not include 0 and their Wald test *p* value <0.05. To understand the biological magnitude of relevant predictors, we used the ggpredict function available in the "ggeffects" package in R v 3.6.1[50,51] to graphically display their model predicted estimates and associated confidence intervals across the range of response variables collected during the study. We fitted all the models using the glmmTMB function in the "glmmTMB" package in R[52]. We calculated AICc$_w$ using the akaike.weights function in "qpcr" package in R[53]. To assess model performance, we plotted residuals against longitude and generated spatial corellograms to investigate the potential for spatial autocorrelation using the "ncf" package in R (Supplementary Fig. 1)[54]. Pseudo-$R^2$ values were calculated for each model in the 95% confidence sets using the "r.squaredGLMM" function in the "MuMIn" R package and based on methods described in ref. [55]. Two sites were omitted from the analysis due to small sample sizes.

Separately, we plotted vector forage ratios for commonly observed non-muroid rodents as a function of the relative probability of python presence (high, moderate, or low), based upon output values of the species distribution models in order to illustrate how host preference changes at different levels of python presence, as preferred non-rodent species, might act as dilution hosts of EVEV where they are still present. Forage ratios are presented as a ratio of species utilization by the vector (fraction of the total blood meal sample) over their rates of availability (relative abundance in the non-rodent community). Although originally developed to quantify food selection by fish[56], forage ratios have been used extensively to quantify host selection by mosquitoes[57].

**Reporting Summary**. Further information on research design is available in the Nature Research Reporting Summary linked to this article.

## Data availability

All relevant data supporting the findings of this study are within the paper and its Supplementary Files. Everglades virus sequences are available in GenBank, submission ID 2465804. Any further data or information are available from the corresponding author upon reasonable request.

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

## Acknowledgements

We thank Ruiyu Pu, Dhani Prakoso, and Sarah Beachboard for supporting molecular work (Everglades virus screening). Hilda Lynn, Carol Thomas, Bethany McGregor, Kristin Sloyer, Agustin Quaglia, and Dinesh Erram helped with mosquito sorting. McKenzie Stewart assisted with mammal trapping. For facilitating research permits and selecting sampling sites we thank Ben Nottingham and Mark Danaher (Florida Panther National Wildlife Refuge); Steve Schulze (Big Cypress National Preserve); and Steve Houseknecht, Mike Owen, Karen Relish, and Skip Fisher (Fakahatchee Strand Preserve State Park). This work was supported by UF DSR Opportunity Fund P0042304 and NIFA FLA-VME-005446.

## Author contributions

N.D.B.-C., E.M.B., M.T.L., and R.A.M. conceived the study and designed experiments. E.M.B., A.A.L., M.C.V., L.E.R., and I.B. carried out field and laboratory work. L.P.C., N.D.B.-C., and R.A.M. analyzed data. All authors provided input and edits on the manuscript and approved the final version.

## Competing interests

The authors declare no competing interests.
