## [Peer Review File · Communications Biology]

Reviewers' comments:

Reviewer #1 (Remarks to the Author):

Invasive Burmese pythons alter host use and virus infection in the vector of a zoonotic virus

Burkett-Cadena et al.

Overview:

The authors test if Burmese python invasion can alter mammal host communities and thus impact the levels of the Everglades virus in mosquitoes. The paper was interesting and tests an important hypothesis.

Major comments:

1. The spatial estimates of Burmese python density is a critical component of these analyses. I had three questions/suggestions in this regard: (a) Is there any direct evidence of differences in python densities between the sites (e.g. camera trap data etc.); (b) I am not a fan of kernel density estimates for presence-only data as they assume a homogenous landscape. I would suggest the authors construct an SDM (e.g., using ensemble approaches as implemented in the R package `biomod2`) and obtain predicted python densities from the ensemble model; (c) As the authors point out the use of citizen science data is associated with biases. One major issue is the spatial bias in effort (e.g., increased efforts near roads or human habitation). These biases pose special problems if the underlying variables are also associated with other variables used in the analyses (e.g., if proximity to roads and human habitation are also associated with higher murid abundance). Several options have been proposed to deal with bias in sampling effort (e.g., Elith et al. 2011 Diversity and Distributions; Stolar and Nielsen 2015 Diversity and Distributions).
2. Throughout the manuscript “rodents:non-rodents” is used but the “non-rodents” seem to include squirrels and rabbits. Maybe “muroid:non-muroid” is a better terminology? I’m also confused about the selection of the species in each category. If cotton rats are the only competent host for the virus, I feel the entire analysis should be done comparing “cotton rat to non-cotton rat” rather than “rodents:non-rodents”.
3. Throughout the manuscript it is difficult to make out the statistical strength of the evidence being presented. If the authors are against reporting *P*-values on their own they can report an alternative measure of statistical support (e.g., effect size etc. as suggested earlier by Halsey 2019 Biology Letters)

Minor comments:

1. Need to check grammar and spelling (e.g., correct spelling of “lcoations”)
2. Why was diversity used for “Non-rodents” but species richness for “Total mammals”?
3. Table 1: Round to same precision for the entire table. Please add model average estimates and standard error (or confidence intervals). It would be good to add some model performance metrics (e.g., R^2 , sensitivity/specificity and/or AUC).
4. Fig. 2: This is a very difficult figure to understand. I would suggest that the data points be sized by sample size. The pie charts be eliminated from within the figure. Also, generally it has been shown that it is easier for readers to visualize proportions in bar vs. pie charts). The bar charts can be plotted outside the main graph area and arrow pullouts can be used to connect the data points to particular bar charts. Alternatively data points could have numbers and these numbers be used to identify the bar charts. Fig 2C: Can the data points here also be scaled by sample size?
5. Fig. 3: Are all the error bars standard error of mean? If this is so it looks like rodent blood meals (Fig. 3b) and EVEV infection rates (Fig. 3c) do not differ significantly between the high and low python

regions. Fig. 3d) The forage ratio according to the reference cited is calculated as “the ratio of the percentage which this same organism makes up of the total population of organisms in the fishes’ environment”. I was thus curious as to how this variable can be negative. Additionally, forage ratios of cotton rats seem to be higher in low vs. high python areas. This seems counterintuitive to the message of the paper. It is possible that a better explanation of the “forage ratio” parameter would help address this confusion (especially because the journal cited as a source is not common to my knowledge).

6. Supplementary figure: Plot so overlapping symbols can be visualized (jitter or semitransparent fill). Add a legend to the figure so it is easy to understand it.

Reviewer #2 (Remarks to the Author):

Invasive Burmese pythons alter host use and virus infection in the vector of a zoonotic virus

By: Burkett-Kadena et al.

This is an original, interesting, and ambitious study evaluating the indirect effect of an invasive predator, the Burmese python on the host composition of mosquito bloodmeals and on the transmission of the Everglades Virus between rodent hosts and, potentially, humans. The hypothesis relies on the python established effect on mammal diversity and the correlation of the mammal composition with the bloodmeal composition of *Culex cedecei*, the only (known) vector of Everglades Virus. To assess this complex hypothesis, the authors mapped the density of the python and selected 12 sites in 3 clusters, one cluster representing high density and the others - low density. In each site, the authors estimated the mammal composition and compared it to the host composition in bloodmeals of *Cx. cedecei*. The results support a reduction in mammal diversity in the cluster of high python density, and an increase in the rodent predominance there. Concomitantly, the authors have measured an increase in the rodent fraction among blood meals of *Cx. cedecei* in the area of high python density. Finally, the authors measured higher virus infection rate of the vector in the cluster with higher python density.

Overall, I enjoyed reading this paper, as it tackles very difficult hypotheses and provides empirical support for a scenario that I thought would be almost impossible to assess. The authors take advantage of a unique system of a virus with a single vector species and only 1-2 reservoir hosts (based on current knowledge), which they have studied extensively. Although the evidence is not overwhelming, the study is compelling, and it addresses an important and timely issue. Therefore, I recommend it for publication. However, I had difficulties in following parts of the Results and the Discussion, and I recommend a revision aimed mostly, at clarification. I believe the paper could be improved by considering the following points.

1. It would help to better visualize the data. For example, I cannot see how many cotton rats (among other rodents) were trapped (in each site) and how many female *Cx. cedecei* were collected, subjected to the molecular assays, and found infected with EVEV in each site. The models used to generate the estimates and effects are complex and difficult to unpack (below), so more visibility of the data will help convey the full picture to the reader.
2. Because the authors have used molecular species identification of the hosts, and only the cotton rat is

the reservoir host for the virus, why not use it instead of the nonrodent to rodent ratio - without diluting it among other rodents, which are not source of infection (as are the medium and large mammals) - to estimate the shift of vector feeding on an infectious host? Using the relative proportion of the cotton rat should replace (or complement) the 'rodent fraction or ratio' especially in the bloodmeal composition (reporting corresponding sample size). Re 1, the authors may add a figure showing the proportion of the cotton rat in the bloodmeals on the X axis and the infection rate of *Cx. cedecei* in the Y axis for each site (12 points), coloring each point by the cluster in different colors by python density (and possibly numbering them, below). I believe such figure would help readers better understand the results as it ties the key elements in a simpler and visually explicit way.

3. Given that cotton rats among other rodents were trapped, were they tested for infection rate?

Although it is not critical, it would be helpful to substantiate a higher prevalence in the vertebrate host in parallel to the vector data.

4. Figure 1: Please show what is the difference in python density between the highest (darkest red) and lowest (yellow) density? Could you show the locations of the observation as dots on the map and how they are related to trails, observation towers, etc.? Please comment whether the observation opportunity confounded this map? What biological factor shapes this distribution?

It'd help to number the sampling sites, so the reader can see the correspondence of the findings to each site and their neighbors throughout. Please explain the rationale for the sampling: why the 12 sites do not evenly span the full spectrum of the python density? They represent essentially one high and one low python densities. Likewise, the clustered subsampling near location is likely to produce correlated results.

5. The Results section opens with complex statistical models of parameters that are not well defined (below). I can't find in Table 1 the sample sizes and degrees of freedom of the tests. The models seem to consider each site separately, however, there seem to be essentially 3 clusters of sites (each with 4 subsamples that are close to each other). Why not pool the subsamples of each cluster? The average cluster values may better reflect the python density and its effect on the mammal composition. The averages may be subjected to more conventional tests. The relatively large variation of mosquito infection between sites in a cluster may reflect small sample sizes (which are not shown); an issue that may be resolved by pooling the subsamples within a cluster. Moreover, the subsamples within cluster are expected to be correlated and ignoring this, may inflate the degrees of freedom of the analysis. Additionally, it is not clear how the seasonal (month) component was treated.

6. In the Introduction or the Discussion the authors may want to comment on whether the incidence of EVEV in humans (and in cotton rats) has increased since the 1980, as I assume, they would predict.

7. I think that parts of the Discussion over-interpret the results. Likewise, striving for simplification would benefit this paper. For example, the distinction between some of the parameter estimates (some better than others in predicting this or that) that are probably correlated, e.g., non-rodent diversity, vs. rodent component in the total mammal diversity, among others, is not very compelling unless the effect sizes are not overlapping or are pointing to opposite direction, etc. Given the complexity of the system, I urge the authors to focus on the most critical components of their story, namely: python density, cotton rat component in mammal diversity, cotton rat fraction in blood meals, and infection rate in the mosquito vector. Most of the other indices tend to blur the picture.

8. Please include in the Discussion a few sentences on the limitations of your study design and how

would you suggest that future studies on related questions be designed. For example, how can one address alternative factors to the python, such as flooding, fire, invasive plants, etc.

Specific comments

L46: Please give 2-3 examples of the most affected species with relevance and include the estimated reduction in population density.

L48: Please give 1-2 examples of the most relevant cascading effects.

L88. Please explain what is the “95% confidence set”

L89. Please define (and explain) what is the “weighted blood meal ratios” (or add a footnote to the table to define it). What are the 7 models? Are these univariate models? Even if you explain it in the methods in detail, the reader needs a brief explanation to move on.

L92. The distinction between (species) composition and activity needs some explanation. Aren't they highly correlated? Is it common to find that activity of the same species in the same season and “habitat” would vary substantially?

Fig 2b. Please add the percentage of the cotton rat in a number (no decimal point) near each pie chart? If possible, please add the total sample size in parenthesis. Note, scaling the pie chart size to the number of observations is fine yet the lowest value may be 7 or 77... the reader can't tell. Also, even if the lowest is 9, how can the reader judge the next size or even if two are the same?

Fig 2c. It seems that one site in the python high density cluster drives the significance of the difference. Could the authors identify the sites to those on the map (see above about numbering the sites)? What were the mosquito infection sample sizes/site (total females subjected to qPCR and maybe the number of pools)? Finally, it'd be helpful to know the month of collection, and possibly (only if you have data on that) the parity as a proxy of mosquito age, to consider if the higher infection rate is due to a confounding factor such as older mosquitoes? These too are not critical, and I only suggest including them if they are available.

L150. I believe you need to remove 'in' before represent

- The term 'cascading effect' is, in my mind, overused, and may be replaced by 'indirect effect'. As the conventional ecological definition is not followed (series of extinctions..), it may be best to define it.

We thank you for thoughtful critique of our work. The editorial board members and reviewers raise important points about measurements of python densities, aspects of species distribution modelling, metrics used to quantify the important aspects of the mammal community (i.e. the focus on cotton rat versus other animals), statistics and data representation in figures. We will address each of these in close detail in the response to each reviewer comment, but I will summarize the major points for your convenience here.

1. We reanalyze the python distribution models using the species distribution models (BioMod2 approach) recommended by reviewer 1, and interpret our findings in light of that approach.
2. We add relevant discussion that directly addresses methods of assessing python abundance to the manuscript.
3. We have re-analyzed our data, focusing on the ratio of cotton rats to all other species, as recommended by both reviewers.
4. All figures have been remade, following requests by reviewers to display the data so that general patterns are recognizable.

We sincerely feel that the changes to the manuscript, based upon the insights and recommendations of the reviewers and editors have greatly improved the quality of the manuscript. We thank you the Reviewers and Editors for their time and contribution.

Reviewer(s)' Comments to Author:

Reviewer 1

Comments to the Author(s)

The authors test if Burmese python invasion can alter mammal host communities and thus impact the levels of the Everglades virus in mosquitoes. The paper was interesting and tests an important hypothesis.

We thank Reviewer 1 for this positive assessment.

The spatial estimates of Burmese python density is a critical component of these analyses. I had three questions/suggestions in this regard: (a) Is there any direct evidence of differences in python densities between the sites (e.g. camera trap data etc.);

(b) I am not a fan of kernel density estimates for presence-only data as they assume a homogenous landscape. I would suggest the authors construct an SDM (e.g., using

ensemble approaches as implemented in the R package biomod2) and obtain predicted python densities from the ensemble model;

Thank you for pointing out a limitation of kernel density estimates. Although the purpose of using this approach was to provide a descriptive comparison of estimated python densities across study sites, we recognize the potential for a homogeneous landscape to interfere with a biologically realistic interpolation. We have now generated a species distribution model (SDM) using the Biomod2 ensemble approach as requested by Reviewer 1. This species distribution model uses python occurrence records, and urban land cover, python home range values, and latitude and longitude raster data as environmental variables. Because sampling bias commonly occurs in such presence-only surveys, and because of a greater number of reports along roadways, we implemented a bias-correction approach to model calibration. The output map of the SDM is presented in Figure 1 and the estimated python activity from the SDM is provided in a summary table (Table 2), by site, along with summary data for key biological variables.

Models were run with default settings, and the ensemble output included GLM, XX, artificial neural networks, g. While the unit of SDM outputs is not a density value per se, the results can be interpreted as a relative probability of presence or in some cases a relative occurrence rate.

Figure 3 shows the relationship between the python relative probability of presence estimates from the SDM and non-rodent activity and host use of the vector, as requested.

(c) As the authors point out the use of citizen science data is associated with biases. One major issue is the spatial bias in effort (e.g., increased efforts near roads or human habitation). These biases pose special problems if the underlying variables are also associated with other variables used in the analyses (e.g., if proximity to roads and human habitation are also associated with higher murid abundance). Several options have been proposed to deal with bias in sampling effort (e.g., Elith et al. 2011 Diversity and Distributions; Stolar and Nielsen 2015 Diversity and Distributions).

We agree that sampling bias is an important consideration when generating species distribution models using citizen science occurrences. SDMs rely on background points, sometimes referred to as 'pseudoabsences', during model calibration. These background points are usually generated randomly by the SDM, but when if sampling bias is present in the occurrence data, it is necessary to adjust the generation of the background data to reflect the

sampling bias of the occurrence data. Here, we restricted the area where background data points could be generated to reflect a similar bias to the citizen science observations, and we calibrated our SDM using a bias-corrected approach.

2. Throughout the manuscript “rodents:non-rodents” is used but the “non-rodents” seem to include squirrels and rabbits. Maybe “muroid:non-muroid” is a better terminology? I’m also confused about the selection of the species in each category. If cotton rats are the only competent host for the virus, I feel the entire analysis should be done comparing “cotton rat to non-cotton rat” rather than “rodents:non-rodents”.

Thank you for pointing out the need for clarification regarding the selection of species in each category and the suggestion to be more direction in our analysis comparing cotton rat to non-cotton rat, rather than rodents to non-rodents. The original category of rodent:non-rodent reflected an effort to be consistent when analyzing communities across two trapping methods (Sherman traps for rodents only, camera traps for all other mammals). We agree with your suggestion that our analysis would be less convoluted, if we focused only on cotton rat: non-cotton rat and have changed the blood meal analysis response variable to the ratio of cotton rat : non-cotton rat. Additionally, we have removed ratios of rodents to non-rodents as explanatory variables because, as stated by the reviewers, it dilutes the analyses. We decided to retain the non-rodent diversity and evenness measures, which are really a measure of camera trap diversity and evenness. The camera traps do not include cotton rat observations, but provide information about other mammal species (deer, raccoons, panthers, bears, etc) at the study sites. We retain a total activity variable that summarizes activity across the Sherman traps and the camera traps, as does total richness. We refrained from calculating a total diversity or total evenness measure to avoid calculating these values across trap types.

3. Throughout the manuscript it is difficult to make out the statistical strength of the evidence being presented. If the authors are against reporting P-values on their own they can report an alternative measure of statistical support (e.g., effect size etc. as suggested earlier by Halsey 2019 Biology Letters)

Thank you for pointing out the need to better clarify the statistical strength of evidence presented in our manuscript. Here, we use a model selection approach, evaluated using an information criterion approach (AICc adjusted for small sample sizes) outlined in detail in Burnham and Anderson 2002. As the reviewer is likely familiar, models are ranked from lowest to highest AICc,

and AICc weights are calculated for each model. We calculated a cumulative sum AICc weights with a threshold equal to 95% to determine whether a model was contributing information. In addition, and as outlined in Halsey 2019 Biology Letters, we summed the AICc weights for each variable included in the 95% confidence set. The cumulative sum of the AICc weights for individual variables range between 0 and 1, with values close to 0 indicating low or no variable importance, and values closer to 1 indicating strong support for the variable. In this regard, although a variable may be in the 95% confidence set of models, it is not a guarantee that the data will support a strong effect of the predictor variable on the response variable. In addition to this model evaluation approach, we investigated model residuals and calculated effect sizes implemented in the 'DHARMA' R package. We reduced the number of variables included in our models because of some redundancy across variables, and now present the effect sizes for these variables.

Minor comments:

1. Need to check grammar and spelling (e.g., correct spelling of "lcoations")

Spelling and grammar have been checked throughout. The spelling of locations has been corrected.

2. Why was diversity used for "Non-rodents" but species richness for "Total mammals"?

We refrained from calculating a total diversity or total evenness measure to avoid calculating these values across trapping methods. We have now removed the species richness variable because we determined that it introduced confusion and in some regards, redundancy with the other variables included in the analysis.

3. Table 1: Round to same precision for the entire table. Please add model average estimates and standard error (or confidence intervals). It would be good to add some model performance metrics (e.g., R2, sensitivity/specificity and/or AUC).

Model average estimates are included in the first row of Table 1. Standard errors are not usually presented in this table because we observing across the model set. We have now included a model diagnostics section, which includes effect sizes with confidence intervals for each model in the 95% confidence set, and we provide additional information regarding model fit. Because of the shift to a generalized linear model framework from traditional least

squares approaches, R2 does not offer accurate information about the model performance or explanatory power. In our case, the parameter estimates may be significant, as present in the DHARMA output, but we cannot describe these relationships as one would using a least squares approach. We feel the addition of the diagnostic plots, effect sizes showing the strength of individual variables, and inclusion of confidence intervals provides a transparent assessment of the model outputs.

4. Fig. 2: This is a very difficult figure to understand. I would suggest that the data points be sized by sample size. The pie charts be eliminated from within the figure. Also, generally it has been shown that it is easier for readers to visualize proportions in bar vs. pie charts). The bar charts can be plotted outside the main graph area and arrow pullouts can be used to connect the data points to particular bar charts. Alternatively data points could have numbers and these numbers be used to identify the bar charts. Fig 2C: Can the data points here also be scaled by sample size?

This figure (now figure 3) has been remade to accommodate the requests of this reviewer. We have added bar charts below the scatter plot to show both the sample size and the species of animals detected across sites, as a function of estimated python density. We attempted to use lines to link the points in scatter plots and the bar charts but the resulting figures were very messy, with crossing lines that disrupted the major emphasis of the charts. However, because the scatterplots follow a left-to-right sequence, so with minimal effort, readers should be able to associate points (given the sample size differences) with their respective bar charts. We hope that the reviewers will be find this satisfactory.

5. Fig. 3: Are all the error bars standard error of mean? If this is so it looks like rodent blood meals (Fig. 3b) and EVEV infection rates (Fig. 3c) do not differ significantly between the high and low python regions. Fig. 3d) The forage ratio according to the reference cited is calculated as "the ratio of the percentage which this same organism makes up of the total population of organisms in the fishes' environment". I was thus curious as to how this variable can be negative. Additionally, forage ratios of cotton rats seem to be higher in low vs. high python areas. This seems counterintuitive to the message of the paper. It is possible that a better explanation of the "forage ratio" parameter would help address this confusion (especially because the journal cited as a source is not common to my knowledge).

Due to other requests from both Reviewers to better display the patterns and relationships between variables in our figures, requests to use cotton rat activity data (instead of all rodents), requests to perform a different spatial analysis of python activity we have remade Figure 3. The new version of

Figure 3 presents the blood meals of cotton rats, relative to other animals (panel 3-C), scaled by sample size. We also present the numbers of bloodmeals for each major group of hosts as bar charts (panel 3-C). The virus infection rate data are presented as a function cotton rat activity (Figure 2-C) and the ratio of blood meals from cotton rat : other animals (Figure 2-D), as requested. Reviewer 1 makes a valid point regarding the forage ratios of cotton rats relative to other rodents. To avoid confusion we have now focused the forage ratio figure on nonrodents (Figure 4), as exploring potential dilution hosts in areas with low python activity was the primary objective of that specific analysis.

6. Supplementary figure: Plot so overlapping symbols can be visualized (jitter or semitransparent fill). Add a legend to the figure so it is easy to understand it.

In the updated version of the supplementary figure points do not overlap.

Reviewer 2

Comments to the Author(s)

This is an original, interesting, and ambitious study evaluating the indirect effect of an invasive predator, the Burmese python on the host composition of mosquito bloodmeals and on the transmission of the Everglades Virus between rodent hosts and, potentially, humans. The hypothesis relies on the python established effect on mammal diversity and the correlation of the mammal composition with the bloodmeal composition of *Culex cedecei*, the only (known) vector of Everglades Virus. To assess this complex hypothesis, the authors mapped the density of the python and selected 12 sites in 3 clusters, one cluster representing high density and the others - low density. In each site, the authors estimated the mammal composition and compared it to the host composition in bloodmeals of *Cx. cedecei*. The results support a reduction in mammal diversity in the cluster of high python density, and an increase in the rodent predominance there. Concomitantly, the authors have measured an increase in the rodent fraction among blood meals of *Cx. cedecei* in the area of high python density. Finally, the authors measured higher virus infection rate of the vector in the cluster with higher python density. Overall, I enjoyed reading this paper, as it tackles very difficult hypotheses and provides empirical support for a scenario that I thought would be almost impossible to assess. The authors take advantage of a unique system of a virus with a single

vector species and only 1-2 reservoir hosts (based on current knowledge), which they have studied extensively. Although the evidence is not overwhelming, the study is compelling, and it addresses an important and timely issue. Therefore, I recommend it for publication.

However, I had difficulties in following parts of the Results and the Discussion, and I recommend a revision aimed mostly, at clarification. I believe the paper could be improved by considering the following points.

We thank Referee 2 for the positive assessment of our work. We hope that we have satisfactorily clarified the outstanding issues in the manuscript, which are summarized on a point-by-point basis below.

1. It would help to better visualize the data. For example, I cannot see how many cotton rats (among other rodents) were trapped (in each site) and how many female *Cx. cedecei* were collected, subjected to the molecular assays, and found infected with EVEV in each site. The models used to generate the estimates and effects are complex and difficult to unpack (below), so more visibility of the data will help convey the full picture to the reader.

To better visualize the data we have remade all figures, attempting to better display sample sizes and relationships between variables. We also include a new table which provides a summary of the data by site.

2. Because the authors have used molecular species identification of the hosts, and only the cotton rat is the reservoir host for the virus, why not use it instead of the nonrodent to rodent ratio - without diluting it among other rodents, which are not source of infection (as are the medium and large mammals) – to estimate the shift of vector feeding on an infectious host? Using the relative proportion of the cotton rat should replace (or complement) the 'rodent fraction or ratio' especially in the bloodmeal composition (reporting corresponding sample size). Re 1, the authors may add a figure showing the proportion of the cotton rat in the bloodmeals on the X axis and the infection rate of *Cx cedecei* in the Y axis for each site (12 points), coloring each point by the cluster in different colors by python density (and possibly numbering them, below). I believe such figure would help readers better understand the results as it ties the key elements in a simpler and visually explicit way.

Thank you for pointing out the need for clarification regarding the selection of species in each category and the suggestion to be more direct in our analysis comparing cotton rat to non-cotton rat, rather than rodents to non-rodents. With this in mind, we reduced the number of variables included in

our candidate model set... The original category of rodent:non-rodent reflected an effort to be consistent when analyzing communities across two trapping methods (Sherman traps for rodents only, camera traps for all other mammals). We agree with your suggestion that our analysis would be less convoluted, if we focused only on cotton rat: non-cotton rat and have changed the blood meal analysis response variable to the ratio of cotton rat : non-cotton rat. Additionally, we have removed ratios of rodents to non-rodents as explanatory variables because, as stated, it dilutes the analyses. We decided to retain the non-rodent diversity and evenness measures, which are really a measure of camera trap diversity and evenness.

3. Given that cotton rats among other rodents were trapped, were they tested for infection rate? Although it is not critical, it would be helpful to substantiate a higher prevalence in the vertebrate host in parallel to the vector data.

This is a valid question, however we did not draw blood samples and so could not determine infection rates. Infection rates in animals is typically very low. The only field attempts to isolate this virus from cotton rats in nature yielded very low infection rates (<1 in 50 animals). Since our trapping effort yielded less than 50 total cotton rats, it is unlikely that infection rates would be very informative. We agree that demonstrating higher infection rates in cotton rats would help to substantiate a higher prevalence in parallel to the vector data and this should be a goal of future studies.

4. Figure 1: Please show what is the difference in python density between the highest (darkest red) and lowest (yellow) density? Could you show the locations of the observation as dots on the map and how they are related to trails, observation towers, etc.? Please comment whether the observation opportunity confounded this map? What biological factor shapes this distribution? It'd help to number the sampling sites, so the reader can see the correspondence of the findings to each site and their neighbors throughout. Please explain the rationale for the sampling: why the 12 sites do not evenly span the full spectrum of the python density? They represent essentially one high and one low python densities. Likewise, the clustered subsampling near location is likely to produce correlated results.

We have updated the kernel density estimation map of pythons and now include a species distribution model to predict relative probability of occurrence across the study area. The SDM was generated using an ensemble modeling approach and executed in the 'biomod2' package in R. Sampling bias is an important component to consider when generating an SDM, and here, we used a bias-corrected approach to ensure that sampling bias from

citizen science occurrence records along roadways or in easily accessible areas did not impact model results. The map included in the revised version represents values between 0 and 1000, with higher values predicting a higher relative probability of occurrence. Details of this modeling approach are provided in the methods section and the supplementary materials of the manuscript.

5. The Results section opens with complex statistical models of parameters that are not well defined (below). I can't find in Table 1 the sample sizes and degrees of freedom of the tests. The models seem to consider each site separately, however, there seem to be essentially 3 clusters of sites (each with 4 subsamples that are close to each other). Why not pool the subsamples of each cluster? The average cluster values may better reflect the python density and its effect on the mammal composition. The averages may be subjected to more conventional tests. The relatively large variation of mosquito infection between sites in a cluster may reflect small sample sizes (which are not shown); an issue that may be resolved by pooling the subsamples within a cluster. Moreover, the subsamples within cluster are expected to be correlated and ignoring this, may inflate the degrees of freedom of the analysis. Additionally, it is not clear how the seasonal (month) component was treated.

We acknowledge the concerns of Reviewer 2 regarding the complex statistical models and the appeal of clustering the sites according to their proximity. Figure 3, panels A, B and C represents our attempt to bridge the divide between the mixed models (the complex statistical models) and conventional tests. Rather than grouping into 3 clusters, as this Reviewer suggested, we had grouped sites into two clusters based upon high and low python activity. We did not subject the means presented in that figure (non-rodent activity, rodent host use, virus infection rate) to parametric statistical comparisons because we felt that the results of general linear mixed model analysis should not be compromised by different tests.

We have now presented the blood meal, virus infection and forage ratios in terms of estimated python presence, as requested by both Reviewers. Data figure 3 and 4 show each of these important metrics in terms of estimated python presence. Importantly, the python presence values do NOT cluster within each larger study area (park), based upon the outputs of the species distribution model.

6. In the Introduction or the Discussion the authors may want to comment on whether the incidence of EVEV in humans (and in cotton rats) has increased since the 1980, as I assume, they would predict.

Reviewer 2 raises an excellent question. Has EVEV incidence increased over the past few decades? Because EVEV is not a reportable disease in the US, there is no database on which to base such assumptions. We would not like to speculate on actual infection rates without data.

7. I think that parts of the Discussion over-interpret the results. Likewise, striving for simplification would benefit this paper. For example, the distinction between some of the parameter estimates (some better than others in predicting this or that') that are probably correlated, e.g., non-rodent diversity, vs. rodent component in the total mammal diversity, among others, is not very compelling unless the effect sizes are not overlapping or are pointing to opposite direction, etc. Given the complexity of the system, I urge the authors to focus on the most critical components of their story, namely: python density, cotton rat component in mammal diversity, cotton rat fraction in blood meals, and infection rate in the mosquito vector. Most of the other indices tend to blur the picture.

Thank you for this constructive comment. We agree that several of the variables included in the candidate model set provided redundant information that would require a nuanced interpretation of the results. With this in mind, we have reduced the number of variables included in our candidate model set and we changed the blood meal ratio to cotton rat blood meals : all other blood meals, which we agree provides a more direct evaluation of our research question. Additionally, we clarify that the cotton rats (and other rodents) were captured using Sherman traps and that all other mammals were recorded using camera traps. Importantly, the camera traps did not record any rodent activity. With this in mind, we include as our predictor variables cotton rat activity (Sherman traps), other mammal activity, except rodents (camera traps), other mammal diversity (camera traps) and cotton rat blood meals : all other animal blood meals in the models investigating EVEV positive mosquito pools.

Models that included the cotton rat blood meals : all other blood meals as the response variable, included cotton rat activity (Sherman traps), other mammal activity (except squirrels, as recorded by camera traps), and other mammal diversity (camera traps).

8. Please include in the Discussion a few sentences on the limitations of your study design and how would you suggest that future studies on related questions be designed. For example, how can one address alternative factors to the python, such as flooding, fire, invasive plants, etc.

We have included a sentence to capture these sentiments. The sentence reads "Future studies might gain a better estimate of human risk using sentinel rodents and sampling from a greater number of locations, and by including additional variables which are known to influence animal communities (flooding, fire, invasive plants, etc.)."

Specific comments

L46: Please give 2-3 examples of the most affected species with relevance and include the estimated reduction in population density.

The sentence now reads "and has been incriminated in precipitous declines in native mammals throughout southernmost Florida [13, 14, 15, 16], with 85-100% decrease in the frequency of observations of raccoon, opossum, bobcat and rabbits [13]."

L48: Please give 1-2 examples of the most relevant cascading effects.

The sentence now reads "The loss of mammal diversity is thought to be causing a complete restructuring of the food web, declines in ecosystem function, and an array of cascading ecological effects [14, 17], such as increased predation on nests of oviparous animals [17]."

L88. Please explain what is the "95% confidence set"

This phrase no longer occurs, as we have rewritten this section from a biological (rather than statistical) perspective.

L89. Please define (and explain) what is the "weighted blood meal ratios" (or add a footnote to the table to define it). What are the 7 models? Are these univariate models? Even if you explain it in the methods in detail, the reader needs a brief explanation to move on.

The term "weighted blood meal ratios" no longer appears in the manuscript. This was an error, and should have been "relative cotton rat host use". This term is defined in Table 3.

L92. The distinction between (species) composition and activity needs some explanation. Aren't they highly correlated? Is it common to find that activity of the same species in the same season and "habitat" would vary substantially?

Well-stated. As several of the variables included in the candidate model set consisted of redundant information, we reduced the number of variables included in our candidate model set.

Fig 2b. Please add the percentage of the cotton rat in a number (no decimal point) near each pie chart? If possible, please add the total sample size in parenthesis. Note, scaling the pie chart size to the number of observations is fine yet the lowest value may be 7 or 77... the reader can't tell. Also, even if the lowest is 9, how can the reader judge the next size or even if two are the same?

The sample sizes are now evident in the lower panels of Figure 3, as these correspond to the actual numbers of rodents detected and blood meals analyzed for each site.

Fig 2c. It seems that one site in the python high density cluster drives the significance of the difference. Could the authors identify the sites to those on the map (see above about numbering the sites)? What were the mosquito infection sample sizes/site (total females subjected to qPCR and maybe the number of pools)? Finally, it'd be helpful to know the month of collection, and possibly (only if you have data on that) the parity as a proxy of mosquito age, to consider if the higher infection rate is due to a confounding factor such as older mosquitoes? These too are not critical, and I only suggest including them if they are available.

L150. I believe you need to remove 'in' before represent

Figure 2 now presents how python presence is distributed across the different variables. Our original presentation of this data did cluster the sites by estimated high and low python "activity" but the new species distribution model for Burmese python, resulted in python presence values that did not cluster within a research area (park). Therefore, we felt it more appropriate to show how patterns of mammal activity, and host use and virus infection varied across the spectrum of predicted python presence, as now presented in Figures 2, 3 and 4.

We did not dissect female mosquitoes to determine age of the population. This is a worthy metric of the vector population, as Reviewer 2 points out, and an important component of vectorial capacity. However, this technique is very time-consuming and was not a part of our study.

We removed "in" before "represent".

- The term 'cascading effect' is, in my mind, overused, and may be replaced by 'indirect effect'. As the conventional ecological definition is not followed (series of extinctions..), it may be best to define it.

We feel that the term “cascading” does have application in our work, as the indirect effects (higher virus infection in the vector) occur as a result of indirect impacts (changes in host use) which occur in response to changes in the mammal community (direct impact). However, we may have indeed overused “cascading”. As such, we changed “cascading” to indirect “in three instances”.

REVIEWERS' COMMENTS:

Reviewer #1 (Remarks to the Author):

The authors have addressed all my comments on the revised manuscript. I have no further comments.

Reviewer #2 (Remarks to the Author):

The authors have thoroughly revised their paper based on the comments and addressed most of the points effectively. Below I provide pointers to suggested additional changes, which are rather superficial. I do not insist on these changes and I'm still enthusiastic about the publication of this work with or without these changes.

Two issues can help improve the paper:

1. The authors tackle a very difficult and important problem and their approach and results are compelling, yet, their data are "soft" and the wording could better reflect the remaining uncertainty as well as the need for further testing and validation even that system. Currently the text conveys a sense of a solid resolution. I appreciate the section of limitation at the end. However, it is too late and it does not include all the key issues. Thus, I suggest a more cautious language throughout the text.
2. Despite the new statistical framework used, I suggest using the WALD test P values in the tables to make the inference more accessible to many readers who are used to count on these statistics to assess the confidence in the results,.

Specific points.

Abstract: The wording "Our results demonstrate how an invasive predator can impact wildlife" is too strong and may be better rephrased. Firstly, the study is correlational and at best can show association. Secondly, the results reveal trends that support the hypothesis, but not overwhelmingly so (below). Without diminished enthusiasm for this work, I believe more careful wording adds credibility to the topic and this paper.

L57. "a striking increase.." please add the values i.e., from x% to y% with refs.

L93-4. "showed that relative cotton rat host use increases approximately five-fold across the recorded range of cotton rat activity (Figure 2-a)," It seems based on the 95%CI that the change was not statistically significant, but the sentence is read as it is. I'd suggest ensuring the text includes reference to the plotted CI. For example, "Although not statistically significant ($P > 0.x, \dots$), the relative cotton rat host use increases approximately five-fold across the recorded range of cotton rat activity (Figure 2-a). - The same issue applies to Figs. 2c and 2d. It is stated that models were selected if their factors' 95%CI did not include zero and the WALD tests was significant at the 0.05 level, however, I cannot see that in the 2a, 2c and 2d (even if the intercept is >0 the slope does not appear to be statistically significant at the 0.05 level by the eye). Could this be explained? Possibly, the p values of the key factors be included in Table 1 (rather than in Supp. Material)?

Fig. 2. Please add sample size in/near the bubbles? I believe 'site level' refers to the nesting of sites with the 3 sampling areas, but this should be defined. How weighting was done is also unclear.

L142-4. "Based on the predictions from our most parsimonious model, we found that as relative cotton

rat host use increased from 0 to > 0.50, EVEV infection rates increased approximately three-fold (Figure 2-c).” - It might be worthwhile to qualify this by mentioning that the change between 0 to 0.4 was minimal and that the effect is based on one site (out of 10). This is still biologically relevant, but mentioning this recognizes the uncertainty in these data.

Table 1 is difficult to comprehend for people who are not familiar with the new statistical analysis/approach used by the authors. Without typical P values of the factors of the model, I’m lost to know how strong the statistical support for the factors listed is (compounded by very broad CI in corresponding figures I mentioned above). To help “old fashion” readers like me, I’d suggest adding asterisks near factor name as conventionally done (* = $p < 0.05$, ** = $P < 0.01$, etc.) to better appreciate the differences among models.

Fig. 4 is confusing because it shows that when species relative activity is high, e.g., white tailed deer then blood meal fraction is low – contrary to my expectation. Even, for the Eastern Cottontail, the second putative dilution host, there is only one point with highest activity and intermediate biting rate. The shaded area is host preference, but it is the same for all species, rather than species specific. I’m probably missing something, but this should be better explained.

Discussion: there is excessive repetition of the key points.

I hope these comments are helpful.

Tovi Lehmann

REVIEWERS' COMMENTS:

Reviewer #1 (Remarks to the Author):

The authors have addressed all my comments on the revised manuscript. I have no further comments.

We thank Reviewer 1 for this positive assessment.

Reviewer #2 (Remarks to the Author):

The authors have thoroughly revised their paper based on the comments and addressed most of the points effectively. Below I provide pointers to suggested additional changes, which are rather superficial. I do not insist on these changes and I'm still enthusiastic about the publication of this work with or without these changes.

We thank Reviewer 2 for their positive assessment and further recommendations.

Two issues can help improve the paper:

1. The authors tackle a very difficult and important problem and their approach and results are compelling, yet, their data are "soft" and the wording could better reflect the remaining uncertainty as well as the need for further testing and validation even that system. Currently the text conveys a sense of a solid resolution. I appreciate the section of limitation at the end. However, it is too late and it does not include all the key issues. Thus, I suggest a more cautious language throughout the text.

We soften the wording throughout the manuscript to better reflect the correlational / associational nature of the data and to highlight the need for further testing and validation of the principal relationships in other systems.

2. Despite the new statistical framework used, I suggest using the WALD test P values in the tables to make the inference more accessible to many readers who are used to count on these statistics to assess the confidence in the results,.

We respect the concerns of Reviewer 2 regarding the reporting of Wald statistic and P values in our results, as this is at the heart of an ongoing (and often heated) debate on the appropriateness of two general statistical frameworks for reporting the results of statistical tests. Thank you for the suggestion of including the Wald p-values when presenting model results. We prefer to present the results in their current format for two reasons. First, we used the glmmTMB package to calculate weighted binomial generalized linear mixed effects models and Wald p-values are not an available option in the model outputs. Our understanding is that reasoning behind this omission is because of the challenges in calculating degrees of freedom when random effects are included in model calibration, including inconsistencies in

calculations across software packages and a lack of consensus on how best to calculate degrees of freedoms when random effects are included. Secondly, the Wald test can be sensitive to boundary effects, which are not uncommon when overdispersion is present, which was the case for our data, and prompted our decision to move to the more complex mixed modeling framework.

Specific points.

Abstract: The wording "Our results demonstrate how an invasive predator can impact wildlife" is too strong and may be better rephrased. Firstly, the study is correlational and at best can show association. Secondly, the results reveal trends that support the hypothesis, but not overwhelmingly so (below). Without diminished enthusiasm for this work, I believe more careful wording adds credibility to the topic and this paper.

We tone down the wording in the abstract. The revised text now reads "...we show that increasing diversity of dilution host (non-rodent mammals) is associated with decreasing blood meals on amplifying hosts (cotton rats), and that increasing cotton rat host use is associated with increasing EVEV infection in vector mosquitoes."

L57. "a striking increase.." please add the values i.e., from x% to y% with refs.

We add requested the percentages and the citations.

L93-4. "showed that relative cotton rat host use increases approximately five-fold across the recorded range of cotton rat activity (Figure 2-a)," It seems based on the 95%CI that the change was not statistically significant, but the sentence is read as it is. I'd suggest ensuring the text includes reference to the plotted CI. For example, "Although not statistically significant ($P > 0.x, \dots$), the relative cotton rat host use increases approximately five-fold across the recorded range of cotton rat activity (Figure 2-a).

The phrase was rewritten to reflect these suggestions. It now reads "Although uncertainty was high at model extremes, model predicted estimates suggest that relative cotton rat host use increases approximately five-fold across the recorded range of cotton rat activity (Figure 2-a.)"

- The same issue applies to Figs. 2c and 2d. It is stated that models were selected if their factors' 95%CI did not include zero and the WALD tests was significant at the 0.05 level, however, I cannot see that in the 2a, 2c and 2d (even if the intercept is >0 the slope does not appear to be statistically significant at the 0.05 level by the eye).

Could this be explained? Possibly, the p values of the key factors be included in Table 1 (rather than in Supp. Material)?

Reviewer 2 raises good points about the 95% confidence intervals in Figures 2a-c. We likely cannot address this issue with our current data. Additional sampling to produce more points and tighter confidence intervals around points will likely be necessary to resolve this question.

Fig. 2. Please add sample size in/near the bubbles? I believe 'site level' refers to the nesting of sites with the 3 sampling areas, but this should be defined. How weighting was done is also unclear.

We have added sample sizes for all points (bubbles) on the figures.

L142-4. "Based on the predictions from our most parsimonious model, we found that as relative cotton rat host use increased from 0 to > 0.50, EVEV infection rates increased approximately three-fold (Figure 2-c)." - It might be worthwhile to qualify this by mentioning that the change between 0 to 0.4 was minimal and that the effect is based on one site (out of 10). This is still biologically relevant, but mentioning this recognizes the uncertainty in these data.

We have rewritten the section to address the reviewer concerns. The passage now reads "Based on the predictions from our most parsimonious model, we found that as relative cotton rat host use increased from 0 to > 0.50 of blood meals, EVEV infection rates increased approximately three-fold (Figure 2-c). High uncertainty in EVEV infection rates was observed at model extremes (Figures 2-c, 2-d), and only one site had very high cotton rat host use (63.6%) and high EVEV infection rate (3.2 / 1,000), which likely influenced model outcomes."

Table 1 is difficult to comprehend for people who are not familiar with the new statistical analysis/approach used by the authors. Without typical P values of the factors of the model, I'm lost to know how strong the statistical support for the factors listed is (compounded by very broad CI in corresponding figures I mentioned above). To help "old fashion" readers like me, I'd suggest adding asterisks near factor name as conventionally done (* = $p < 0.05$, ** = $P < 0.01$, etc.) to better appreciate the differences among models.

While we certainly respect the concerns of Reviewer 2 regarding the reporting of P values in our results (I also learned stats in the old fashioned manner), we have elected to present the results in their current format for two reasons. First, we used the glmmTMB package to calculate weighted binomial

generalized linear mixed effects models and Wald p-values are not an available option in the model outputs. Our understanding is that reasoning behind this omission is because of the challenges in calculating degrees of freedom when random effects are included in model calibration, including inconsistencies in calculations across software packages and a lack of consensus on how best to calculate degrees of freedoms when random effects are included. Secondly, the Wald test can be sensitive to boundary effects, which are not uncommon when overdispersion is present, which was the case for our data, and prompted our decision to move to the more complex mixed modeling framework.

Fig. 4 is confusing because it shows that when species relative activity is high, e.g., white tailed deer then blood meal fraction is low – contrary to my expectation. Even, for the Eastern Cottontail, the second putative dilution host, there is only one point with highest activity and intermediate biting rate. The shaded area is host preference, but it is the same for all species, rather than species specific. I'm probably missing something, but this should be better explained.

We recognize that the host preference / forage ratio plots are not easy to interpret. We have attempted to clarify in the Discussion that some hosts were preferred at low and others at intermediate python density. Because each animal species represents a fraction of the total relative abundance of the community, each species effectively "competes" for blood meals, such that it is uncommon to have multiple preferred species at a given site. The modified text states "Marsh rabbit and white-tailed deer were found to be selected by *Cx. cedecei* (Figure 4) where relative probabilities of python presence were low or moderate, respectively."

Discussion: there is excessive repetition of the key points.

We have eliminated some of the repetitious points in the Discussion, as requested.

I hope these comments are helpful.

Tovi Lehmann

The comments are very helpful. Thank you Dr Lehmann.